

# Nature and Extent of Shallow Marine Convection in Subtropical Regions: Detection with airborne and spaceborne Lidar-Systems over the tropical North Atlantic Ocean

Manuel Gutleben[1], Silke Gross[1], Martin Wirth[1], and Andreas Schäfler[1]

[1]Deutsches Zentrum für Luft- und Raumfahrt (DLR), Institut für Physik der Atmosphäre, 82234 Wessling, Germany.

*Correspondence to:* MANUEL GUTLEBEN (manuel.gutleben@dlr.de)

**Abstract.** Shallow marine cumulus convection over the Atlantic ocean near Barbados is studied with observations by airborne and spaceborne lidar instruments performed during the field campaign Next-generation Aircraft Remote Sensing for Validation Studies (NARVAL). For the first time airborne lidar measurements with the DLR high spectral resolution lidar system WALES on-board the German research aircraft HALO were conducted over the tropical North Atlantic Ocean. In the course of NAR-

VAL several CALIPSO satellite underflights were performed, which allow comparisons of detected cloud top edges from the two lidar instruments (i.e. WALES and CALIOP on-board CALIPSO). The study concentrates on the comparison and investigation of detected cloud top height distributions derived from measured WALES and CALIOP lidar profiles by use of a newly developed cloud detection algorithm. This allows to test the utilization of satellite based lidar systems for the observation of shallow marine convection. The distribution of cloud top heights during wintertime measurements shows a two-layer structure

with maxima in ∼1000 m and ∼2500 m in both WALES and CALIOP measurements. Cloud top heights vary with latitude. The analysed WALES and CALIOP data shows most frequent cloud tops in 10° to 20° N at heights from 1500 to 2500 m. A meridional decrease of detected cloud top heights over the subtropical North Atlantic Ocean, with lower values in the North, is observed. Approximately 36% of all clouds in the Atlantic trades are detected to have a horizontal extent of less than 1 km in the winter season. Cloud gaps shorter than 1 km dominate the Atlantic trades. They make up approximately 45% of all detected

cloud gaps.

## 1  Introduction

In present day climate research clouds are one of the major contributors to uncertainties in estimates of the Earth's energy budget (Stephens, 2005). Clouds are highly variable in space and time and their occurrence cannot be predicted exactly. The quantification of cloud feedbacks in climate models is still one of the biggest challenges in present day climate science (Bony

et al., 2015). Inadequate representation of clouds and moist convection in general circulation models is the main limitation in current representations of the climate system (Stevens and Bony, 2013). Large inter-model spread is found in low latitudes due to the dependence of cloud and precipitation responses on unresolved processes (Stevens and Bony, 2013). Differences in the representation of shallow marine convection in global climate models lead to large differences in climate sensitivity estimates (Bony and Dufresne (2005); Zelinka et al. (2012)).





These so called trade wind clouds form at the top of the subtropical marine boundary layer beneath the descending branches of the Hadley cell. They are often limited in altitude to at most 4000 m due to the presence of a prominent trade wind inversion, separating the moister marine boundary layer from the dry free troposphere (Stevens, 2005). Hence, they play an important role for the transport of moisture to the free atmosphere (Tiedtke, 1989). Shallow marine cumuli contribute about 60% to the

net cloud radiative forcing and are one of the dominant contributors to global albedo (Hartmann et al., 1992). They cover about 12% of the sky over the Earth's oceans (Warren et al., 1986), but are extremely variable in spatial extent with time. The importance of shallow marine convection on climate in the trade wind regions was already identified in the mid-20$^{th}$ century when campaigns started to explore the characteristics of shallow marine convection by studying meteorological processes and the structure of the moist boundary layer with the aid of airborne in-situ observations (Malkus, 1954, 1956, 1958). Since then

a large number of field experiments were conducted to better characterize shallow marine trade wind convection for numerical atmospheric models.

Hereby, macrophysical properties of cumulus clouds like their morphology, cloud size distributions or cloud top height (CTH) distributions play an important role. On the one hand these properties are used to evaluate cloud models (e.g. Siebesma and Cuijpers (1995)), on the other hand they serve as input parameters for model calculations to investigate the clouds radiative

and dynamic effects on the environment (e.g. Zhao and Austin (2005a), Zhao and Austin (2005b)). Ground-based, shipborne or airborne measurements during field campaigns provide highly resolved observations of the macro- and microphysical cloud properties (e.g. Colòn-Robles et al. (2006); Siebert et al. (2013)), but are limited in space and time. Satellite measurements provide global coverage and long-term observations but the footprint of passive satellite observations mostly exceeds the small-scale structure of trade wind convection. In contrast, active remote sensing satellite measurements with the Cloud-Aerosol

Lidar with Orthogonal Polarization (CALIOP) on-board the Cloud-Aerosol Lidar and Infrared Pathfinder Satellite Observations (CALIPSO) have a footprint diameter of 70 m and a footprint spacing of about 330 m (Winker et al., 2010). Thus, such measurements have frequently been used to study macrophysical parameters of trade wind cumuli and for model evaluation (e.g. Luo et al. (2016), Medeiros et al. (2010), Ahlgrimm and Köhler (2010), Tackett and Di Girolamo (2009)). However, up to now, no systematic evaluation of the applicability and constraints of CALIOP data for the use in studies of shallow marine

convection was done.

The objective of the presented study is twofold. In a first step we want to evaluate if, and to which extent, spaceborne lidar measurements with the CALIOP lidar can resolve the small-scale character of trade wind convection in comparison to highly resolved airborne measurements. We therefore use measurements performed during the Next-generation Aircraft Remote Sensing for Validation Studies (NARVAL) mission (Klepp et al., 2014) on-board the German high-altitude and long-range research

aircraft HALO (Krautstrunk and Giez, 2012) in December 2013 over the subtropical North Atlantic Ocean in combination with the spaceborne lidar system CALIOP. In a next step we want to use these airborne and spaceborne lidar measurements to investigate the size of shallow marine convection as well as their CTH distribution during NARVAL. Furthermore, we want to investigate the representativeness of the data collected during the NARVAL mission and therefore compare these data sets to longer time periods of CALIOP measurements and of different seasons. Exemplarily we use data of December, January and

February (DJF) 2012/2013 and of summer (June, July, August - JJA) 2013.





A description of the used instruments and their measuring principles is given in Section 2. Section 3 presents the comparison of airborne and spaceborne lidar measurements and the statistical analysis of the NARVAL measurements with respect to the macrophysical properties of shallow marine convection. In Section 4 the results are discussed with respect to representativeness and comparisons to other studies. Section 5 concludes this work.

## 2 Method and instrumentation

### 2.1 NARVAL

In December 2013 and January 2014, the Next-generation Aircraft Remote Sensing for Validation Studies (NARVAL) mission took place. NARVAL was designed as an airborne experiment using the German High Altitude and Long range research aircraft (HALO), which is a modified Gulfstream G550 business jet with a maximum range of more than 12000 km and a maximum cruising altitude of more than 15500 m (Krautstrunk and Giez, 2012). During NARVAL, the HALO aircraft was equipped with a set of remote sensing instruments. Besides the two main instrument packages, the water vapour and differential absorption lidar WALES (Water Vapour Lidar Experiment in Space, Wirth et al. (2009)) and the Halo Microwave Package (HAMP), a combination of a 36 GHz cloud radar and a set of microwave radiometers (Mech et al., 2014), the payload also included a miniDOAS system (Prados-Roman et al. (2011), Weidner et al. (2005)) and radiation measurements (Fricke et al., 2014). In this study we only make use of the lidar data. The NARVAL mission consisted of two separate measurement periods. In December 2013, measurements were performed out of Oberpfaffenhofen to Barbados as transfer flights as well as out of Barbados as local flights. The aim of the December flights was to study shallow marine convection and their environment. Measurements were conducted in trade wind regions over the subtropical North Atlantic between 10 and 20 December 2013 at the beginning of the climatic dry season. In January 2014, HALO was operated out of Keflavik (Iceland) to study post-frontal convection and precipitation over the extra-tropical North Atlantic. In the following, the abbreviation NARVAL only refers to the flights performed during the first phase in December 2013 as this study focuses on the characterization of shallow marine trade wind convection. During NARVAL, eight research flights with almost 70 flight hours were conducted - four Caribbean transfer flights from or to Oberpfaffenhofen (EDMO, Germany) and four local flights over the subtropical North Atlantic Ocean departing from Grantley Adams Airport (TBPB, Barbados). Six CALIPSO underflights were performed over the subtropical North Atlantic Ocean between 25 ° W and 60 ° W. Figure 1 shows the flight tracks of the conducted HALO flights during NARVAL. Furthermore, all CALIPSO ground tracks within a predefined area used for the analysis in this study during the NARVAL period are plotted. Altogether 12 daytime and 13 nighttime CALIPSO tracks crossed the defined measurement area during NARVAL. An overview of the conducted research flights during NARVAL including times of take-off and landing, and times of CALIPSO underflights is given in Table 1.



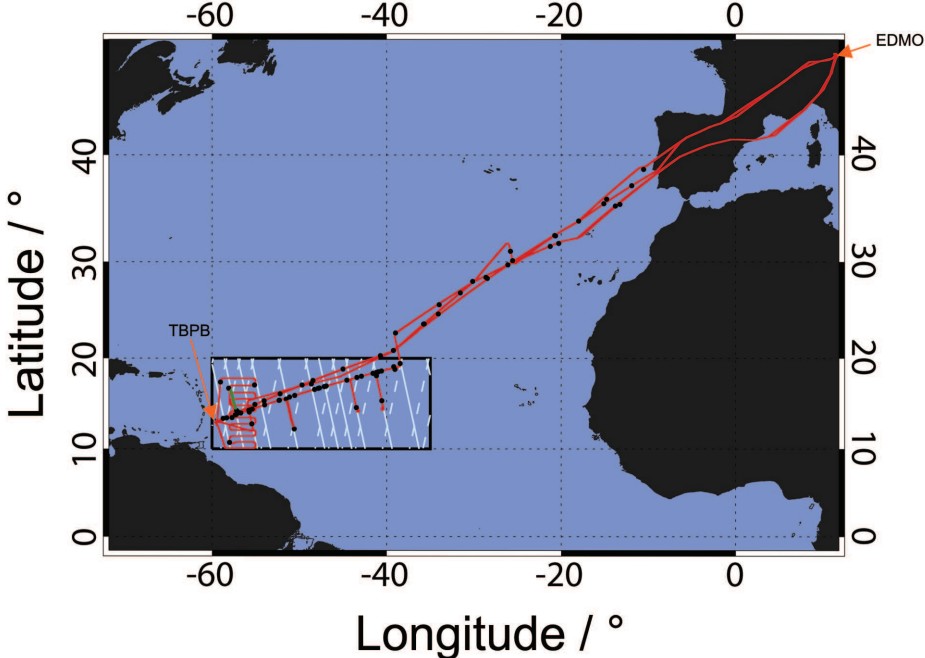

**Figure 1.** Flight-tracks of CALIPSO and HALO during NARVAL. White lines represent CALIPSO flightpaths in the investigation area ranging from $60°W$ to $35°W$ and $10°$ to $20°N$ (period: 10 to 20 December 2013, dashed: descending node, solid: ascending node). Red lines show the tracks of conducted HALO research flights. The green line indicates the profile discussed in Section 3.1.1. Black dots mark locations of dropped radiosondes.

**Table 1.** Overview of the conducted research flights during NARVAL (times given in UTC).

| Date | Take-off | Underflight | Landing |
|---|---|---|---|
| 10.12.2013 | 10:14 (EDMO) | $15:08$ | 20:41 (TBPB) |
| 11.12.2013 | 14:29 (TBPB) | $17:26$ | 21:58 (TBPB) |
| 12.12.2013 | 13:50 (TBPB) | $16:30$ | 20:20 (TBPB) |
| 14.12.2013 | 13:35 (TBPB) | $16:18$ | 20:21 (TBPB) |
| 15.12.2013 | 15:15 (TBPB) | $17:01$ | 21:45 (TBPB) |
| 16.12.2013 | 13:10 (TBPB) | $16:07$ | 22:59 (EDMO) |
| 19.12.2013 | 10:05 (EDMO) | - | 19:57 (TBPB) |
| 20.12.2013 | 16:20 (TBPB) | $17:19$ | 02:35 (EDMO) |



## 2.2 Lidar systems

For the this study, we use measurements of two different lidar systems - the airborne system WALES and CALIOP onboard the CALIPSO satellite. Both are briefly described in the following.

### 2.2.1 The WALES instrument

WALES is an airborne demonstrator built for a proposed, spaceborne mission to observe $H_2O$ concentrations in the atmosphere from space using the Differential Absorption Lidar (DIAL) technique (Bösenberg, 1998). In addition to the water vapour channels, WALES is equipped with High Spectral Resolution Lidar (HSRL) capability using an iodine filter (Esselborn et al., 2008), allowing simultaneous HSRL measurements at 532 nm and DIAL measurements in the absorption bands of water vapour between 935 and 936 nm. Furthermore, WALES performs polarization sensitive measurements at 532 and 1064 nm.

This study concentrates on the HSRL measurements at 532 nm. The raw data of WALES has a vertical resolution of 15 m and a horizontal resolution of $0.2~s$ ($\sim 40~m$ at $200~ms^{-1}$ cruising speed). The data is post-processed to retrieve the aerosol backscatter ratio and the aerosol backscatter coefficients $\beta_{part}$ (Esselborn et al., 2008). For further technical information and description of the system see Wirth et al. (2009) and Esselborn et al. (2008).

### 2.2.2 The CALIOP instrument

The CALIPSO satellite has a sun synchroneous orbit in an altitude of about $705~km$ in nadir-pointing orientation. It crosses the equator at 13:30 (ascending node) and 01:30 (descending node) local solar time and has a 16-day repeat cycle. CALIOP is the spaceborne lidar instrument (Winker et al., 2007) on-board CALIPSO (Winker et al., 2010). It is a backscatter lidar performing simultaneous polarization sensitive measurements at 532 and 1064 nm. CALIOP data are provided in different data processing levels, from raw data to data products with large horizontal and vertical averaging lengths. For this study we use CALIOP Level

1B V4 532 nm data with a 30 m vertical and 330 m horizontal resolution expressed in terms of total attenuated backscatter coefficient $\beta_{tot}$.

## 2.3 Data evaluation

For a consistent comparison of WALES and CALIOP instruments, data sets are converted into a common unit. In this study we use the ratio between total backscatter coefficient and molecular backscatter coefficient which is called backscatter ratio

($BSR = \beta_{tot}/\beta_{mol}$). To calculate the molecular backscatter coefficient $\beta_{mol}$, temperature and pressure fileds from Integrated Forecasting System model analysis (IFS) of the European Centre for Medium-range Weather Forecasts (ECMWF) are used. The modelled fields are interpolated in space and time to match the flight paths of CALIPSO and HALO and their specific range resolutions.

To determine CTHs based on the $BSR$ data, we define a $BSR$ threshold for the cloud-/no-cloud decision. During NARVAL,

aerosol was mainly located in the marine boundary layer. These aerosol layers showed $BSR$ values between 1 and 10. BSR values for clouds were found to be much higher. For this study we define a threshold of $BSR = 90$ that marks the lower edge



of the $BSR$ range found for clouds during NARVAL. For the examination of shallow marine trade wind convection we only consider height ranges less than 4000 m altitude as shallow marine clouds usually do not advance to greater altitudes (Stevens, 2005). Furthermore, all determined profile heights less than 250 m (all altitudes are given above sea level) are excluded, as they are prone to surface echoes. The cloud detection algorithm scans profiles in the direction of beam propagation. If the threshold

is exceeded once in a single lidar shot, the corresponding height is marked as CTH and the profile is marked as cloudy (Figure 2). We are aware that this detection algorithm does not capture lower cloud layers. However, we think that this approach gives a good upper estimate for the CTH distribution. Considering multiple cloud layers within one profile in the detection can even lead to false conclusions since the lidar signal is mostly saturated within clouds.

In a next step, all cloudy profiles are connected to determine cloud size distributions along the flight-paths. To detect beginnings

and endings of a cloud, two cloudy profiles must be separated by at least one cloud-free profile. Otherwise, they will be attributed to the same cloud.

To compute lengths of clouds and cloudless areas, the Earth's shape is simplified to be a spheroid. The distance $\Psi$ between two points A and B on a surface of a sphere is then calculated using,

$$\Psi = arccos[sin(\theta_A)sin(\theta_B)$$
$$+ cos(\theta_A)cos(\theta_B)cos(\Phi_B - \Phi_A)]., \tag{1}$$

where $\Phi$ is the azimuthal angle and $\Theta$ is the polar angle. For distances of cloud and cloud gap points, $\Psi$ is multiplied with the sum of the radius of the Earth ($r \approx 6370\ km$) and the mean CTH.

## 3   Results

### 3.1   CALIOP / WALES comparison

To evaluate the potential of satellite based lidar measurements with the CALIOP lidar on-board CALIPSO we proceed as

follows. First, we perform direct comparisons of the measured profiles during the CALIPSO underflights including the comparison of the derived CTH distribution. In a next step, we compare the macrophysical properties from airborne and spaceborne lidar for the whole NARVAL measurement period.

**Case study - 11 December 2013**

During all NARVAL flights, the overall synoptic situation stayed constant. Figure 3 gives an exemplary overview of the situa-

tion on 11 December 2013. Small irregularly scattered clouds dominate the area over Barbados and over the Atlantic Ocean. South of about 10° N deep convective structures from the inter-tropical convergence zone (ITCZ) are present and north of about 30° N cloud structures of extra-tropical weather regimes are visible. On 11 December 2013, we sampled the airmasses west of Barbados in several east-west flights at different latitudes form about 18° N to about 10° N. Along most parts of the flight path the situation was characterized by a moist marine sub-cloud layer below about 1 km topped by small scale cloud

structures at $\sim 1\ km$ and at $\sim 2.5\ km$. Weak aerosol structures were also visible between these two cloud layers. Above about




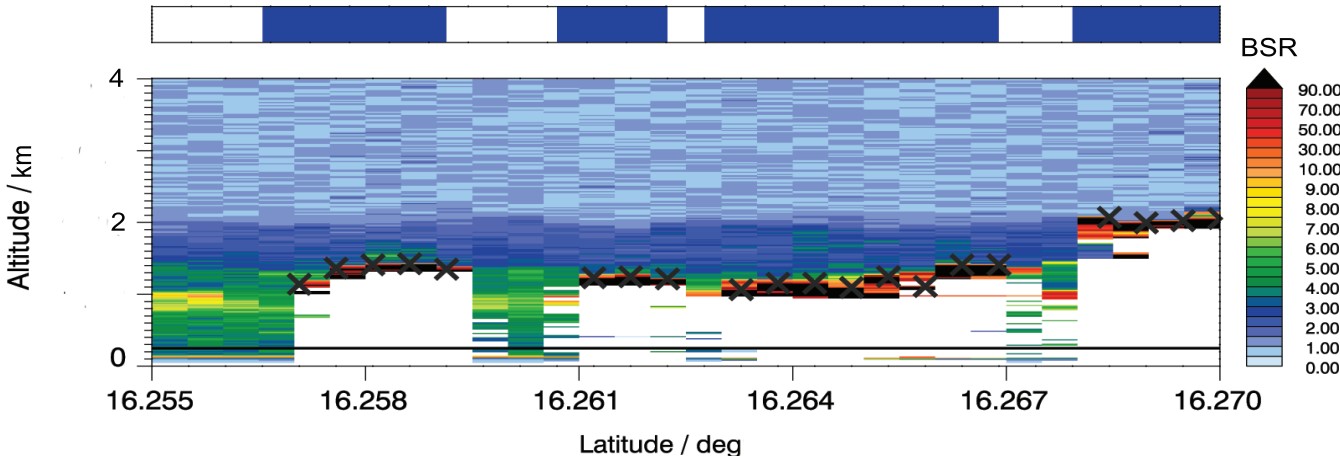

**Figure 2.** Exemplary visualisation of the developed algorithm applied on WALES data along the flight track ($\sim 2\ km$). Colors represent measured backscatter ratios. Grey crosses indicate detected cloud tops. The horizontal solid black line marks the surface echo cut-off. The uppermost panel shows sequences of detected clouds (blue-filled polygons).

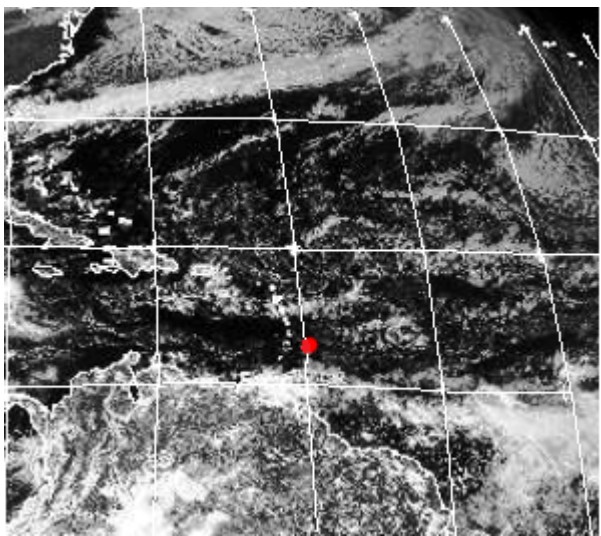

**Figure 3.** GOES - Satellite visible light image showing the large scale cloud situation around Barbados on 11 December 2013, 11 UTC. The red dot indicates the location of the island of Barbados (http://www.goes.noaa.gov).



3 km the atmosphere showed no further aerosol structures. DIAL measurements (not shown here) indicated low water vapour concentration in the free troposphere. South of about 12° N lidar measurements indicated cirrus clouds above 10 km altitude. These cloud structures became more distinct in the southernmost part of the flight track. Comparing the lidar cross-section along the flight track with the satellite images one can see that these structures can be attributed to the ITCZ. During the mea-

5    surement flight, a CALIOP underflight was performed on a flight track between 14.1° N and 57.2° W to 16.9° N and 57.8° W. This underflight is also marked in Figure 1. The $BSR$ cross-section of the WALES lidar and the CALIOP lidar of this about 320 km long flight track is shown in Figure 4 (a,b). CALIPSO passed this flight track in less than one minute. HALO needed

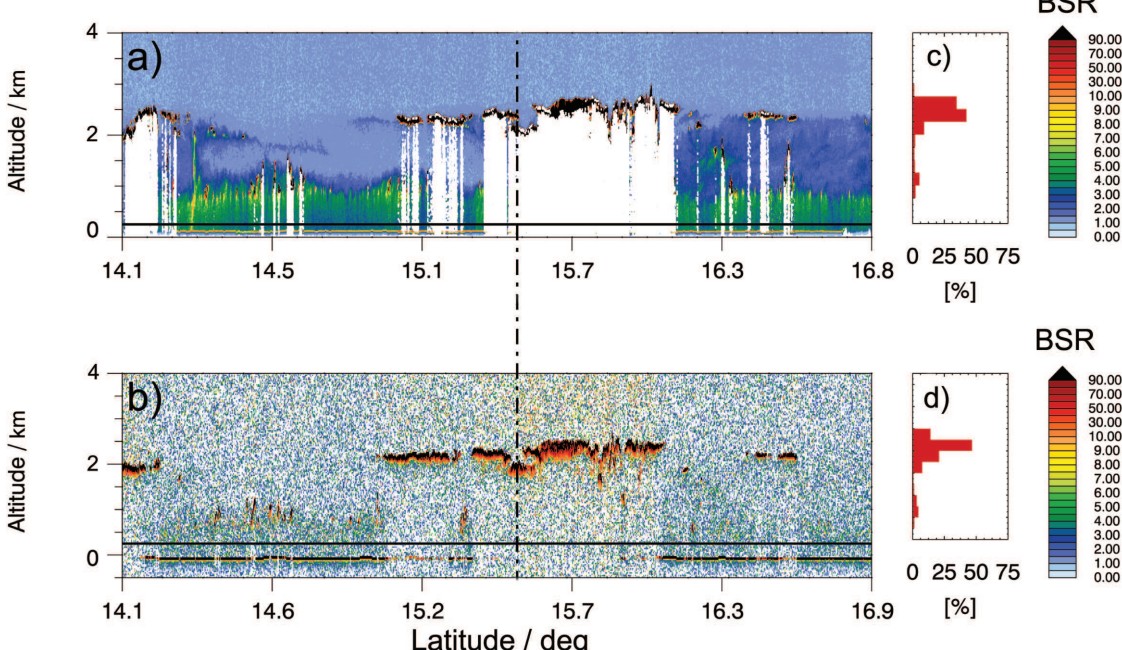

**Figure 4.** Vertical profiles of determined $BSR$ during a CALIPSO underflight on 11 December 2013 (WALES (a) and CALIOP profiles (b)). The cloud top detection algorithm is applied above 250 m (horizontal black solid line). The moment of vertical collinearity is marked by a black dashed line. Cloud top height frequencies are shown in (c) and (d) with 250 m vertical bin-size.

35 minutes (from 17:13 to 17:37 UTC) to sample the same area.

Although the CALIOP measurements have a lower signal-to-noise ratio, the general aerosol and cloud structure looks the same

10   in the WALES and the CALIOP measurements. Differences in the two measured cross-sections are mainly seen at the begin-
ning of the underflight. Due to the different speeds of HALO and CALIOP especially highly variable cloud structures may
have changed at the beginning and end of the track. Between latitudes of 15.1° N and 16.2° N the situation is dominated by a
stratiform-like cloud structure with a horizontal extent of approximately 125 km. Besides this stratiform-like cloud structure
small scale convective clouds with horizontal extents of less than 1 km are present at the top of the marine sub-cloud layer.



Figure 4 (c, d) shows the CTH distribution along the flight track of the underflight derived from WALES and CALIOP measurements. In general, both CTH distributions show a good agreement with a two-layer cloud structure. Both distributions have their maximum in detected CTH frequency at heights between 2250 to 2500 m due to the detected stratiform-like cloud layer. Over 47% of all CTHs derived from CALIOP measurements and 43% of all CTHs derived from WALES measurements

are found in this height range. However, while the CTH distribution from the WALES measurements shows high values in the height bins between 2250-2500 m and 2500-2750 m with about 43% and about 32%, respectively, the CTH distribution derived from CALIOP measurements shows a contribution of almost 50% in the height bin between 2250-2500 m but a significantly lower value above. In contrast it shows about 10% more CTHs located in the height bin between 2000-2250 m compared to the CTH distribution derived from WALES measurements. Both distributions show local maxima in the order of 5% in height

bins between 750 and 1250 m, representing the small scale convective clouds. No CTHs are detected above 3000 m.

## 3.2 Cloud top heights during NARVAL

In a next step we compare the overall distributions of detected CTHs during the whole period of NARVAL. We first analyse the CTH distribution of all conducted CALIPSO underflights and in a next step the CTH distribution of all conducted WALES and CALIOP measurements in the defined research area during NARVAL. Underflight profiles were found by manual filtering.

An overview of the number of used data sets and profile kilometres is given in Table 2.

Figure 5(a) presents the CTH distribution of all conducted underflights. Most of the detected cloud tops are found in heights between 2000 to 2500 m. More than 50% of all detected CTHs derived from measurements of both instruments are found in these heights. As already seen in the case study on 11 December 2013, the CTH distribution derived from WALES measurements shows larger values in the uppermost part of the height range 2000-2500 m while the CALIOP derived distribution

shows a higher contribution in the lower part of this height range. Furthermore, the CTH distribution derived from both instruments have a local maximum in heights between 1000 and 1250 m, with WALES detecting almost twice as many cloud-tops as CALIOP in this height region (WALES: $\sim$ 15%; CALIOP: $\sim$ 7%). The differences in relative frequency of detected CTHs in each bin interval between the two compared data sets never exceed 10%. There are no overall systematic differences in the distributions, although WALES data shows a more pronounced two layer CTH structure. No cloud tops are found in heights

between 2750 and 4000 m.

Figure 5(b) shows the CTH distributions for the whole period of NARVAL in the defined area. As seen in the CTH distribution of the underflights the CTH distributions for all NARVAL measurement flights derived from WALES and CALIOP are in good agreement. Again, a two-layer structure is obvious with maxima in height ranges between 750-1500 and 1750-2500 m. The maximum derived from WALES measurements is found between 2250 and 2500m, while from CALIOP measurements it is

found between 2000 and 2250 m. Altogether, about 60% of all CTH are located in height ranges between 1750 and 2750 m for both measurement systems. In the height range between 750 and 1500 m about 21% and 25% of all detected clouds were found from CALIOP and WALES measurements, respectively.





**Table 2.** Overview of the used data sets during the NARVAL period listing the profile kilometres in the box ranging from $60°W$ to $35°W$ and $10°$ to $20°N$ (period: 10 to 20 December 2013).

|  | Profile km |
| --- | --- |
| Underflights | $\sim 2300\ km$ |
| WALES Data (NARVAL period) | $\sim 32000\ km$ |
| CALIOP Data (NARVAL period) | $\sim 24500\ km$ |

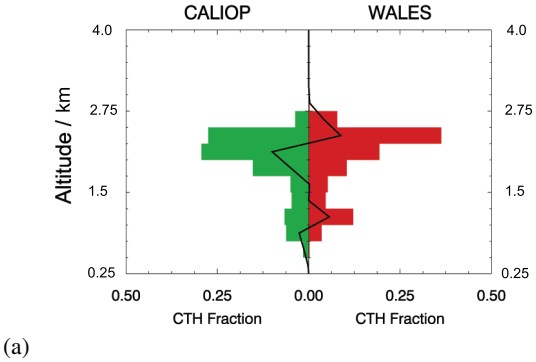

(a)

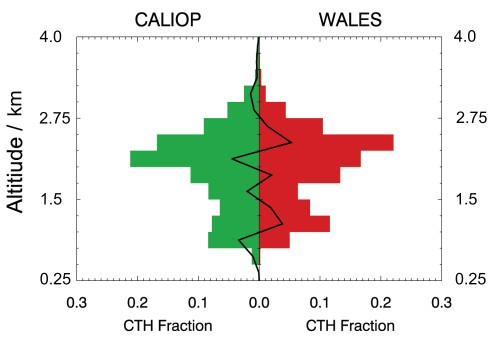

(b)

**Figure 5.** Relative frequency distributions of detected cloud top heights measured by WALES and CALIOP - (a) during all conducted CALIPSO underflights and (b) during the whole NARVAL period (spatial boundaries: 60 and $35°$ W and 10 and $20°$ N). The black lines indicate the differences between the compared profiles for each bin-interval with a bin-size of 250 m.

## 3.3 Meridional distribution of detected cloud top heights

In the previous section we have shown that CALIOP measurements provide a good basis for studying the vertical distribution of CTHs. Consequential, we use the CALIOP measurements to study the meridional distribution of shallow marine convection CTHs to determine possible changes in their distribution with latitude. For this analysis we use CALIOP data over the subtrop-

5  ical North Atlantic Ocean between $60°$ and $35°$ W longitude and between $0°$ and $30°$ N latitude. Figure 6 shows the meridional distribution of the CTHs derived from all CALIPSO overpasses in the defined area during the entire NARVAL period. One can see that low clouds are omnipresent over the entire meridional transect from $0°$ to $30°$ N. Most of the cloud tops are found at heights between 1500 and 2500 m. North of about $20°$ N a lowering of CTHs compared to the regions between $10°$ to $20°$ N where the maximum of the cloud tops is found in higher altitudes, is clearly visible. Between $20°$ and $30°$ N the maximum

10  in CTH frequency decreases from about 2500 m to about 1500 m. This can be explained by differential heating and therefore deepening of the marine boundary layer towards the tropics. Furthermore, an abrupt transition from high CTH frequency to low CTH frequency is located at approximately $10°$ N. This marks a transition to deep convective systems near the ITCZ. High



surface air temperatures and subsequent instability in the tropics are preconditions for air parcels to overcome a trade wind inversion and to build up deep convection.

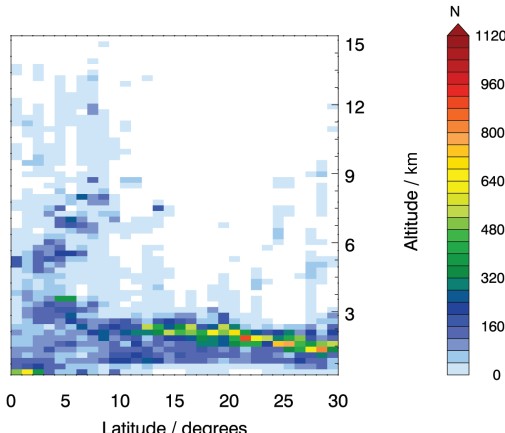

**Figure 6.** Meridional distribution of detected relative single shot cloud top height frequency (10 to 20 December 2013) with height plotted in a two-dimensional histogram - meridional binsize: $1°$, vertical binsize 250 m, bounding longitudes: $60°$ and $35°$ W.

## 3.4 Cloud lengths and cloud gap lengths

For the analysis of cloud lengths, clouds with clearly detected boundaries are used (see Section 2.3). Figure 7 shows the
5  two-dimensional frequency distribution of cloud lengths combined with mean CTHs. During NARVAL shallow marine clouds with a horizontal extent of less than 1 km were prevalent. They make up approximately 38% in CALIOP measurements. Also stratus clouds with extents larger than $100\ km$ are detected in 7% of all CALIOP measurements. The maximum in mean CTH is located between 2000 and 2250 m. Most of the clouds have a size less than 1 km and CTHs between 2500 and 2750 m. They make up about 9% of all detected clouds. WALES data (not shown) and CALIOP are in good agreement, showing nearly the
10  same distributions of relative cloud length frequency within this 10-day period. However, due to the better horizontal resolution of the WALES data, we were also able to resolve cloud sizes smaller than 1 km (up to 0.5 km - not shown). We found a uniform distribution of cloud sizes between 0 and 1 km. This resolution is not possible with CALIOP measurements as they have an effective resolution of about 330 m.

Figure 8 illustrates the length distribution of cloudless areas along the CALIOP flight paths observed during NARVAL. Most
15  gap lengths have a length of less than 1 km. They make up nearly 60% in analysed WALES data (not shown) and about 45%



in analysed CALIOP data. The frequency declines with the length of cloud gaps in an exponential behaviour. Detected lengths which are longer than 10 km make up approximately 10%.

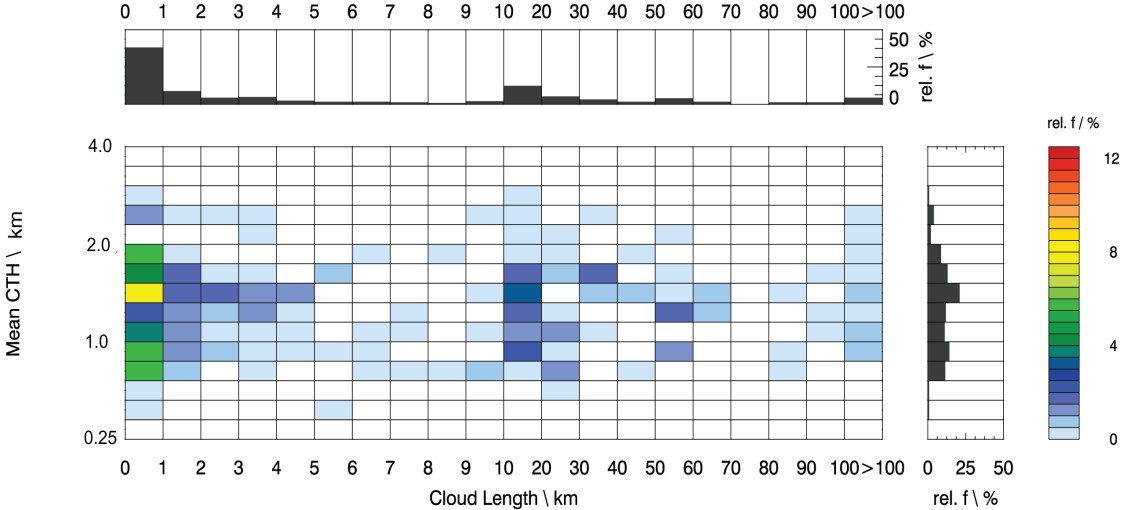

**Figure 7.** Relative frequency of detected mean cloud top heights and associated cloud lengths from CALIOP measurements during NAR-VAL (10 to 20 December 2013), visualized by a two-dimensional histogram (bin interval [height bins]: 250 m, bin-interval [length bins]: logarithmic, spatial boundaries: 60° to 35° W and 10° to 20° N.)

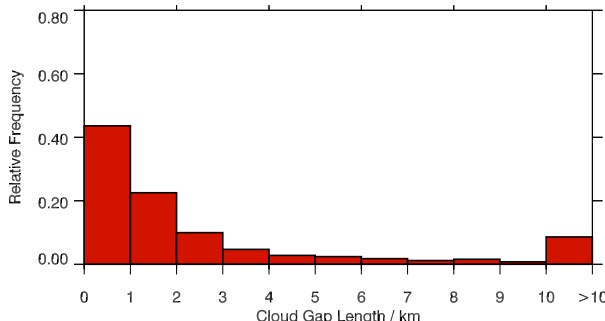

**Figure 8.** Relative distribution of cloud gap lengths of CALIOP measurements (10 to 20 December 2013) in longitudes of 60° to 35° W and latitudes of 10° to 20° N.



## 4   Discussion

During NARVAL, a 10-day data set was collected. To determine if the data set sampled in this short period of time is representative for longer periods, we also analyse CALIOP data measured in December, January and February (DJF) 2012/2013 at the beginning of the dry season, and in June, July and August (JJA) 2013 at beginning of the wet season, and compare them to the
NARVAL results.

In a first step we compare the CTH distribution derived from CALIOP measurements with the CTH distributions for DJF 2012/2013 and JJA 2013 (Figure 9). The distribution of JJA 2013 looks different compared to the NARVAL result and also in comparison to DJF 2012/2013 distribution. Looking at the CTH distributions during NARVAL and in DJF 2012/2013 we see that both winter periods show a two layer cloud structure. During NARVAL, the maximum with 21% of all cloud tops is
found between 2000 and 2250 m. In DJF 2012/2013, the maximum is found in lower height bins from 1750 to 2000 m with 16% of all cloud tops. Both periods show a local maximum between 750 and 1000 m. While the distribution in lower height ranges from 750 to 1250 m is the same in both CTH distributions the second mode at higher levels is generally lower during DJF 2012/2013 measurements, indicating that general environmental differences may have an influence on cloud heights. Comparing the detected CTH distribution during NARVAL with those derived for the JJA 2013, we see significant differences.
The most pronounced difference is that the CTH distribution during JJA 2013 shows one-modal layering. This is in agreement with the CTH distribution found by Ahlgrimm and Köhler (2010) for July 2006, 2007, and 2008. However, they also looked at a CTH distribution for January 2007 and found no significant differences to their July measurements neither in shape of the distribution nor in height of the most frequent CTH range. During the JJA 2012 period, we find a maximum between 1250 and 1500 m. These differences between winter and summer months lead to the assumption that there might be seasonality in the
vertical distribution of trade wind clouds.

Comparing the extent of clouds we found no significant differences in cloud size and cloud gap size (not shown here). 37% of all detected clouds in DJF 2012/2013 data show lengths less than 1 km. In JJA 2013 the number of clouds with sizes smaller 1 km increases to 42%. Both values agree well with the portion of about 45% found during NARVAL.

The meridional CTH distribution of DJF 2012/2013 and the one captured during NARVAL show the same features (Figure
10 (a)). A lowering of the cloud tops north of about 20° N and signatures of the ITCZ south of 10° N are clearly visible. The meridional CTH distribution derived from CALIOP data in JJA 2013 looks slightly different (Figure 10 (b)). It shows a maximum between 15° and 20° N with CTH around 1500 m ASL. North of this layer the number of detected CTH decreases. Overall less cloud tops of trade wind convection are detected in JJA 2013 compared to DJF 2012/2013. However, more cloud tops in heights of 6000 to 9000 m are detected in the trade wind regions in JJA 2013. This might be an indication that more
deep convective systems are present in summer months.

## 5   Summary

To investigate shallow marine cumulus convection, the NARVAL mission took place over the tropical North Atlantic Ocean in the vicinity of Barbados in December 2013. This study analyses lidar measurements performed during NARVAL, in com-



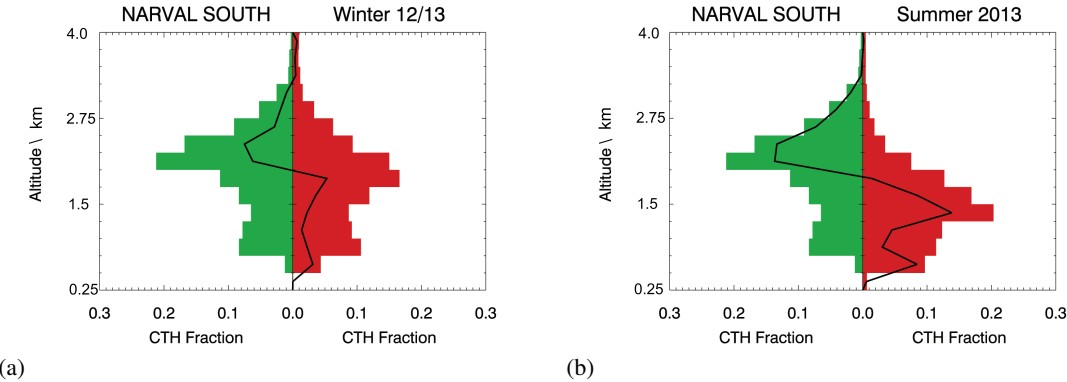

(a)                                                             (b)

**Figure 9.** Relative distributions of detected single-shot cloud top heights measured by CALIOP. The black line indicates the differences between the compared profiles for each bin-interval with a bin size of 250 m. (a) Comparison of detected cloud top heights during the period of NARVAL (10 to 20 Dec 2013) and winter 2012/2013 (December, January, February). (b) Comparison of detected cloud top heights during the period of NARVAL (10 to 20 Dec 2013) and summer 2012/2013 (June, July, August) - spatial boundaries: 60° and 35° W and 10° and 20° N.

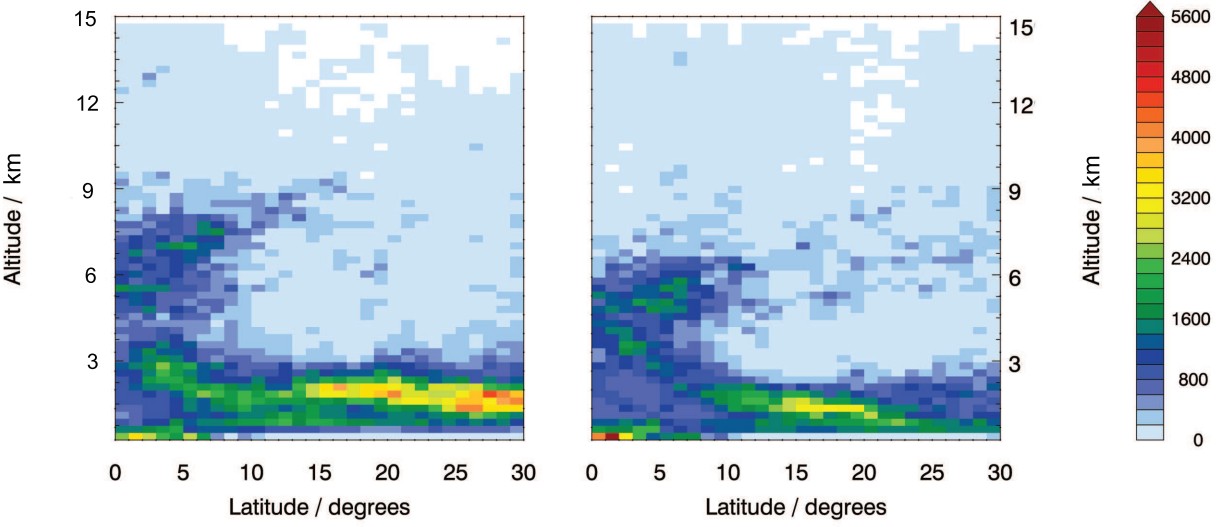

**Figure 10.** Meridional distributions of detected absolute cloud top height frequencies (DJF 2012/13 (left) and JJA 2013 (right)); Meridional bin-size: 1°, vertical bin-size: 250 m, bounding longitudes: 60° and 35° W.





bination with spaceborne CALIOP measurements. Comparisons of WALES and CALIOP measurements performed during six daytime underflights show, that beside the lower signal to noise ratio CALIOP measurements are in principle well suited to study shallow marine convection, although very fine structures of clouds cannot be resolved due to the resolution of the CALIOP measurements. Overall, CTH distributions derived from CALIOP and WALES measurements are in good agreement.

During NARVAL it is found that the CTH distribution of trade wind clouds show a two-layer structure. This two-layer structure is also found for CTH distribution derived from CALIOP measurements in DJF 2012/2013. However, CTH distributions from CALIOP measurements in JJA 2013 show a one-modal structure in height, leading to the assumption that there might be seasonality in the distribution and occurrence of trade wind clouds. Furthermore, it is found that about $45\%$ of all clouds have a length smaller 1 km. But also stratus clouds are detected with cloud lengths of $> 100$ km which make up $\sim 7\%$ of all clouds.

Shallow marine cumulus convection shows a meridional variability. NARVAL results are well representative for the conditions of winter seasons and the dry season respectively. Measurements in DJF 2012/2013 show the same meridional distribution as found during NARVAL.

In this study the reasons of the difference in CTH and meridional distributions between winter and summer measurements were not examined. Differences can be caused by variabilities in the general circulation pattern or due to the influence of aerosols.

Aerosols play an important role for the development and lifetime of clouds (Twomey, 1977) and may also modify the stability of the atmosphere (Gasteiger et al., 2016). Aerosol transport over the Atlantic Ocean and thus the aerosol distribution in the trade wind region is highly dependent on the general circulation and is subject to seasonal variations (Kaufman et al., 2005).

In August 2016, further airborne measurements in the vicinity of Barbados were conducted during the NARVAL-II mission to study shallow marine trade wind convection and exchange processes from shallow to deep convection. These measure-

ments provide the opportunity (together with this study) to further examine the seasonality of trade wind clouds, the general circulation pattern as well as the effect of aerosols on cloud development.

*Acknowledgements.* The authors like to thank the staff members of the DLR HALO aircraft from DLR Flight Experiments for preparing and performing the measurement flights. All CALIPSO data were obtained from NASA Langley Research Center Atmospheric Science Data Center. Profiles of atmospheric parameters, e.g. pressure and temperature, were provided by ECMWF. The NARVAL mission was funded

by the Max-Planck-Institute of Meteorology, the University of Hamburg, the Deutsches Zentrum für Luft- und Raumfahrt (DLR), and by the Deutsche Forschungsgemeinschaft (DFG). This study was funded by a DLR VO-R young investigator group within the Institute of Atmospheric Physics.




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
