# Peer review of "Nature and Extent of Shallow Marine Convection in Subtropical Regions: Detection with airborne and spaceborne Lidar-Systems over the tropical North Atlantic Ocean"

_Atmospheric Measurement Techniques, 2016_

## Referee Comment (RC1) · Anonymous Referee #1 · 14 Dec 2016

**"Nature and extent of shallow marine convection in subtropical regions: detection with airborne and space borne lidar-systems over the tropical north atlantic ocean"**
by Gutleben, M., Gross, S, Wirth, M. and Schaefler, A.

**General comments and recommendation**

This manuscript compares cloud top height and cloud size distributions in a typical trade-wind cumulus region from measurements by a lidar aboard the HALO aircraft and aboard the CALIPSO satellite. I was very excited to read the manuscript, because it exploits new lidar measurements to look at a type of clouds that is important for understanding climate. I was hoping to read either a discussion on the technical capabilities and limitations of either lidar system at accurately detecting these clouds, or learn something new about the distribution of these clouds when viewed from above - with some of the best instruments around. But the manuscript largely disappointed me.

What does the manuscript want to achieve? Given that this is submitted to a journal that focuses on a discussion of atmospheric measurement techniques I had expected to read much more details about the actual measurement and analysis methods. But details on why the authors have chosen specific thresholds on backscatter ratio's, or how the molecular and total backscatter are calculated, are missing. The authors also do not discuss how the different lidars and their footprints explain measured differences in CTH distributions, even if small. The fact that lidars measure clouds only in the along-track direction, effectively missing information on the 2D dimensions of clouds, and thus possibly leading to a bias in cloud length or gap distribution, is never even mentioned.

I also did not gain any new physical insight from this manuscript, which largely glances over a rich body of literature that has looked at the distribution of cloud layers from these clouds (from ground-based measurements e.g., most notably the Barbados Cloud Observatory, but also from space-borne sensors e.g., MISR, ASTER), and which does not consider another rich body of literature that has described the evolution of cloud types and cloud depth along trade-wind trajectories. The discussion and summary that touch upon influences of the aerosol on the change in cloud top heights as one moves towards the equator feels misplaced.

A lack of instrumental details and technical insights on the one hand, and on the other hand, a lack of interpretation and putting the measurements into perspective of existing work, has led to my decision to reject the manuscript. Below I have outlined a few specific points that motivated and might further clarify my decision.

- The abstract describes that this paper will test the utilization of CALIOP at observed trade-wind cumuli, yet there is no sentence that specifically states how well CALIOP performs, and which remaining biases in CALIOP data - if any - remain. The abstract ends with two sentences that discuss cloud gaps, and which have no specific meaning when the authors do not explain what their findings on cloud gaps implies for cloud cover, for the heterogeneity of the cloud field, or for something else. These sentences refer to findings that are discussed with as much as two sentences in the main body of the paper. Is that worthy of ending the abstract?

- The introduction: P2L24: what do the authors understand by systematic evaluation? And in L26-35: How is step 1 of this systematic evaluation different from step 2? Furthermore, what is meant with "small-scale" character (L27)? This could be cloud droplet number concentrations, dimensions of clouds, cloud cover, irregularity of cloud edges, etcetera.

- Section 2.3: P5: How are the total and molecular backscatter defined? I could not repeat the author's analysis from this description alone. How variable is the molecular component, e.g., is it absolute necessary to account for this? If so, what is the order of magnitude with which this varies? How do the authors motivate their BSR thresholds? L31 reads: "BSR values for clouds were found to be much higher". What independent source did the authors use to note that at such values they are dealing with clouds?

- Section 2.4: P6: How relevant is taking into account the spherical nature of the Earth when most clouds tend to be smaller than 1 km? Or, how many long clouds did you detect that made you choose to take the sphericality into account, and what error would you have made if not doing so?

- Section 3.1: P6 and the satellite image in Figure 3. L25 read: "small irregularly scatter clouds dominate the area over Barbados and the Atlantic Ocean". When I look at the satellite image, I actually observe a large number of very large clouds clustered together in specific areas, and I see very few small cumuli. One should either zoom in to see the clouds you are talking about, or change the interpretation of this image.

- Section 3.1: P8L14: the stratiform cloud in Figure 4 does appear to have some gaps in between - is this really 125 km and counted as one cloud?

- Section 3.2: Can you put your findings into perspective of recent studies looking at similar statistics but using ground-based lidars at the Barbados Cloud Observatory? What are the downsides of flying at high altitudes and looking down, compared to being on the ground and looking up? What are you missing?

- Section 3.3 and Figure 6: There are many studies that have looked at the change in cloud properties along trade-wind trajectories and which have done very detailed work on trying to decipher what controls cloud break-up and cloud deepening when moving closer to the Equator. Think of all the studies that discuss stratocumulus to cumulus transitions, and the GPCP transition cases.

- Section 3.4 and Figure 7: This figure clearly shows that (unlike suggested in preceding text) that when cloud top heights are detected near 2 - 2.5 km that they are as often associated with individual small but deep clouds as they are associated with extended stratiform-like layers.

- Discussion and Summary: Any differences you find between different periods are of course related to changing synoptic conditions and changing seasons, hence, those should no longer be just assumptions. Several other studies using both ground-based and satellite data have demonstrated a seasonality in trade-wind cumuli in the region, as well as explained what meteorological factors (and not the aerosol) are responsible.

---

## Referee Comment (RC2) · Anonymous Referee #2 · 2 Jan 2017

Major comments It is a great idea to compare the cloud observations from spaceborne CALIPSO and airborne measurement. As the authors stated in introduction, a systematic evaluation of the CALIPSO data set is desirable. But the abstract fails to clarify the important comparison results. "Cloud top heights vary with altitude" is common sense. Introduction is of clarifying the scientific or practical necessaries, not a pile of references. The introduction in the manuscript is not well organized and cannot reflect the importance of this study. I suggest the authors reorganize and rewrite the introduction with a focus on what this study can contribute. 1. I cannot get the main idea of the manuscript from the title. What doses "nature" in the title stand for? The title should be

revised to properly summarize the manuscript. 2. The structure of the introduction is not clear. For example, the first two paragraphs are just a list of references without a clear conclusion from them or logical structure. What do the authors want to say with those two paragraphs? 3. The manuscript strongly suffers from its poor written English and fails to efficiently deliver information. I just list some of the grammar errors and improper expressions in minor comments, but the authors need to thoroughly rewrite the manuscript. 4. The section discussion is unnecessary. It discusses some details of previous sections. It is better to recognize them into their corresponding sections. I am interested to this topic, and this study is important for the CALIPSO applications for the scientific community. According to the above concerns, however, I suggest the authors thoroughly rewrite the manuscript and resubmit.

Minor comments: 1. Erroneous English grammars and expressions a. The abbreviations must be defined before using them in abstract. b. The paragraph in section 2.1 is too long to follow. c. Page 1 lines 6-8: "The study concentrates on the comparison and investigation of detected cloud top height distributions derived from measured WALES and CALIOP lidar profiles by use of a newly developed cloud detection algorithm." d. Page 2 lines 1-3: Not clear. e. Page 2 lines 15-22: What does "footprint" mean? Resolution? f. Page 2 lines 28-31: Delete "We therefore . . . system CALIOP". g. Page 5 line 1: "For the this study" ? h. Page 6 lines 19-20, 24, and 27-30. I cannot understand. i. Please use comma when necessary. Please see page 8, lines 12-14; page 9, lines 8-9; page 15, line 5, . . . j. Page 6 line 24. What does "stayed constant" mean? k. Page 6 lines 26-28: . . . form about 18 to about 10 N . . . ? l. Page 13 line 14: "Comparing" -> comparing with. Please fix this error throughout the text. 2. The reference styles are not consistent (Page 16). 3. Please add labels for Figure 3. 4. The calculation of cloud and cloudiness lengths is common practice, so the equation and its related description are not necessary.

Please also note the supplement to this comment:
http://www.atmos-meas-tech-discuss.net/amt-2016-333/amt-2016-333-RC2-

supplement.pdf

---

## Author Comment (AC1) · 20 Mar 2017

Please find the final response to the reviewer comments (RC1 and RC2) and the revised manuscript in the supplement.

Kind regards, Manuel Gutleben

Please also note the supplement to this comment:
http://www.atmos-meas-tech-discuss.net/amt-2016-333/amt-2016-333-AC1-supplement.zip